# OpenReview forum: "Leveraging Importance Sampling to Detach Alignment Modules from Large Language Models"
_NeurIPS.cc/2025/Conference — NeurIPS 2025 poster_

### Official Review · Reviewer_Nz81 · 2025-06-29

**Clarity:** 3
**Significance:** 3
**Originality:** 3
**Rating:** 5
**Confidence:** 4

**Summary:**

This paper proposes a Residual Alignment Model (RAM), which is based on the residual correction method of alignment but further enhanced by formalizing the alignment process as a type of importance sampling. In this framework, the unaligned model serves as the proposal distribution, and a secondary sampling model the alignment process. Instead of aligning on the full output sequence, importance sampling modifies the LLM output probability on the token level. This method enables decoupled sampling between the model and the aligner, thereby increasing the inference efficiency for alignment.

**Questions:**

In Eqn. 2, what’s f(x)?

**Ethical Concerns:**

["NO or VERY MINOR ethics concerns only"]

**Final Justification:**

Not changed.

**Limitations:**

yes

**Paper Formatting Concerns:**

No.

**Quality:**

3

**Strengths And Weaknesses:**

**Strength**:
1. The importance sampling method enables token-level alignment on the aligner module, which accelerates the inference process for residual-based alignment. The idea is simple and interesting. The proof is mostly clear.
2. Extensive experiments on instruction following, domain adaptation, and preference optimization proved the effectiveness of this method. RAM also significantly accelerates the inference process compared to Aligner.


**Weakness**:
1. In line 105-107, why do you “maximize” the lower bound? From my understanding, the lower bound should be minimized to lower the SFT loss. This is also inconsistent with equation 8, which aims to maximize the aligner log prob over aligned responses in S and minimize on the proposal module, emphasizing learning the residuals of the probability of distributions.

---

> ### Author Rebuttal · Authors · 2025-07-31
>
> Thank you for your insightful comments. We genuinely appreciate your recognition of our work and hope our responses address the concerns raised.
>
> >W1-1: In line 105-107, why do you “maximize” the lower bound? From my understanding, the lower bound should be minimized to lower the SFT loss.
>
> The right side of Equation 6 represents a lower bound of the SFT loss, directly minimizing this lower bound does not effectively minimize the loss itself (unlike minimizing the upper bound). Compared with Equation 4, the primary contributor to reducing the original loss is the negative term $E_{x \sim S, y \sim P_M}[log Q_\theta(y|x)]$. Therefore, to achieve the goal of minimizing the loss, we can initially maximize this term, allowing the right side of Equation 6 to approach the loss as closely as possible, and then minimize the expression to achieve the target loss minimization.
>
> However, the adoption of this optimization approach necessitates addressing a min-max problem (refer to Equation 32 in [1]), complicating the process. Notably, the term $E_{x \sim S, y \sim P_M}[log Q_\theta(y|x)]$ has some upper bound, denoted as $U$. To mitigate the negative contribution of this term, we can introduce a positive term of the form $-\lambda(E_{x \sim S, y \sim P_M}[log Q_\theta(y|x)]-U)$ in Equation 7, This compensation effectively substitutes the maximization of the term $E_{x \sim S, y \sim P_M}[log Q_\theta(y|x)]$, thus formally aligning the loss. By simplifying Equation 7, we derive Equation 8, which encapsulates our objective function for minimization.
>
> [1] Song Y, Kingma DP. How to train your energy-based models. arXiv preprint arXiv:2101.03288. 2021 Jan 9.
>
> >W1-2: This is also inconsistent with equation 8, which aims to maximize the aligner log prob over aligned responses in S and minimize on the proposal module, emphasizing learning the residuals of the probability of distributions.
>
> Upon examining Equation 8, since $\alpha=1−\lambda$, minimizing the loss can be interpreted as minimizing the $E_{x \sim S, y \sim P_M}[log Q_\theta(y|x)]$ by a factor of 1 while simultaneously maximizing it by a factor of $\lambda$. The combined effect of these two actions results in the minimization of the term by a factor of $\alpha$.
>
> I hope this explanation aids in your understanding of our method.
>
> ---
>
> >Q1: In Eqn. 2, what’s f(x)?
>
> The $f(x)$ is any function of interest whose expected value we want to estimate with respect to the target distribution $Q$. This is a common representation of Importance Sampling [1], although the term $f(x)$ is not explicitly referenced in this paper.
>
> [1] Tokdar ST, Kass RE. Importance sampling: a review. Wiley Interdisciplinary Reviews: Computational Statistics. 2010 Jan;2(1):54-60.

---

> > ### Comment · Reviewer_Nz81 · 2025-08-03
> >
> > Thank you for your response. I think my current ratings reflect my final assessment of this paper. Good luck.

---

> > > ### Author Response · Authors · 2025-08-04
> > >
> > > Thank you for your recognition of our work! Wishing you all the best in your work and life!

---

### Official Review · Reviewer_K98W · 2025-07-01

**Clarity:** 3
**Significance:** 2
**Originality:** 2
**Rating:** 4
**Confidence:** 3

**Summary:**

This paper proposes a framework called RAM to make the process of aligning LLMs more efficient and less resource intensive. The proposed method reframes alignment using the principles of importance sampling, separating the model into a large, fixed base model and a smaller, adaptable alignment module. Here, a primary contribution is this detached framework, expressed as a product of the two modules' probabilities, which permits the smaller aligner to be trained independently. The paper also presents an iterative as well as token-level decoding algorithm to address the issue of first-token latency often seen in similar multi-stage models. The effectiveness of the proposed pipeline is demonstrated through experiments on different tasks with varying specifications, including instruction following and preference optimization. It is shown to generally perform better than baseline methods like standard fine-tuning.

**Questions:**

Please see my comments above in Strengths And Weaknesses

**Ethical Concerns:**

["NO or VERY MINOR ethics concerns only"]

**Final Justification:**

I would like to keep my current positive rating since authors' response largely consolidates my initial positive review of their work.

**Limitations:**

yes

**Quality:**

3

**Strengths And Weaknesses:**

Strength:

+ This paper tries to tackle a real and practical problem of making LLM alignment affordable and efficient. The proposed idea is intuitive: using one large base model that can be adapted for many different tasks with small and inexpensive "aligners" seems like a pragmatic and scalable approach for real-world use. To achieve this objective, authors propose to apply importance sampling to formulate the alignment process, where the pretrained frozen model refers to the proposal distribution, while the adapter estimates the importance score.

+ With theoretical insights, authors propose a training pipeline utilizing an efficient sequence-level training objective, by increasing the probability of desired responses and using the proposal distribution as a regularization.

+ Experiments are performed on multiple benchmark datasets


Weakness:

1.	I am a little confused about the statement that “Importance sampling …, which is useful when direct sampling is difficult or of large variance”. Although being unbiased, it is well known that the importance sampling in its original form can be unstable due to the high variance problem. In this case, how will your proposed solution tackle this problem when the target and proposal distribution deviates from each other. Meanwhile, will the unbiasedness introduced by importance sampling matter here? If not, is it possible we can substitute importance sampling with some biased estimators for reduced variance for enhanced training stability?

2.	In the proposed pipeline, it seems that the aligner's role is limited to re-weighting the pre-selected set of candidates, where it cannot introduce new tokens that were not initially proposed. This can create a fundamental bottleneck. If the optimal, aligned response requires a token that the unaligned Proposal Module deems low-probability and thus fails to include in its initial set of proposals, the RAM framework has no mechanism to generate it. The final output is therefore constrained by the quality and diversity of the initial proposals, which actually corresponds to the main motivation of this work of adding an additional aligner module.

3.	If my understanding is correct, one assumption here is that the Proposal Module and the Residual Aligner will share the same vocabulary space, which can lead to restricted application scenarios as well as the plug-n-play nature of the proposed pipeline. In this case, these two modules will need to come from the same model family of the same generation, so that they will be able to share the same vocabulary space. Authors also mention this limitation in their Appendix.

4.	No statistical significance results are provided for the main results, but only for the ablation study. Meanwhile, could authors also include a study in terms of the efficiency. For instance, what will be the running time and inference time overhead for adding the proposed method?

---

> ### Author Rebuttal · Authors · 2025-07-31
>
> Thank you for the insightful comments and helpful suggestions! We hope that our responses help alleviate the concerns that were raised.
>
> ---
>
> >W1: I am a little confused about the statement that “Importance sampling …, which is useful when direct sampling is difficult or of large variance”. Although being unbiased, it is well known that the importance sampling in its original form can be unstable due to the high variance problem. In this case, how will your proposed solution tackle this problem when the target and proposal distribution deviates from each other. Meanwhile, will the unbiasedness introduced by importance sampling matter here? If not, is it possible we can substitute importance sampling with some biased estimators for reduced variance for enhanced training stability?
>
> Thank you for pointing out the issue with the phrase "of large variance" in the context of importance sampling. We intended to convey that "importance sampling itself can increase variance," and we will revise it accordingly.
>
> Now, let's address how we tackle this problem:
>
> In importance sampling, high variance in the importance weights $W=P_S/P_M$ can occur if the proposal distribution $P_M$ poorly approximates the target distribution $P_S$. This is often due to **support mismatch** and **probability mismatch**. Our task, as pre-defined in L67-L69, "utilize instances from the biased **subset** $S$ to adapt the model $P_M$", assuming $P_M$ supports $P_S$, thus focusing only on **probability mismatch**.
>
> The RAM method mitigates this by introducing a learnable residual aligner $Q_\theta$ to adjust $P_M$ for better alignment with $P_S$, reducing mismatch. And then, by normalizing with the partition function $Z_\theta$, RAM ensures that $P_\theta$ is a valid probability distribution. This normalization means the weights $W$ are no longer solely dependent on $P_M$, but are adjusted by $Q_\theta$, which learns corrections for specific regions where $P_M$ deviates from $P_S$, effectively smoothing out large weights and minimizing variance.
>
> Moreover, during inference, RAM samples candidate tokens from `Top-P/Top-K` regions of $P_M$, as you mentioned in `W2`, to maintain high $P_M$ values, leading to relatively low importance weights $W$. Additionally, we employ self-normalized importance sampling in the Proposing-Aligning-Reducing Sampling (L137) to further reduce variance, even though it introduces some bias.
>
> **In summary, we propose strategies during both training and inference to reduce variance and enhance stability with biased estimators.**
>
> ---
>
> >W2: In the proposed pipeline, it seems that the aligner's role is limited to re-weighting the pre-selected set of candidates, where it cannot introduce new tokens that were not initially proposed. This can create a fundamental bottleneck. If the optimal, aligned response requires a token that the unaligned Proposal Module deems low-probability and thus fails to include in its initial set of proposals, the RAM framework has no mechanism to generate it. The final output is therefore constrained by the quality and diversity of the initial proposals, which actually corresponds to the main motivation of this work of adding an additional aligner module.
>
> We fully acknowledge your point regarding the potential oversight of tokens deemed low-probability by the Proposal Module, which may be valuable for alignment in certain contexts. This inevitably leads to an important question: **Is there a singular optimal response for alignment?**
>
> In tasks such as **mathematics and reasoning**, there is often a definitive correct solution. However, given that mathematical and reasoning capabilities are key evaluation metrics for general large models (such as the one used as the Proposal Module), it is unlikely that the optimal response would be categorized as low-probability.
>
> In contrast, **value alignment and domain adaptation** tasks frequently address subjective preferences. For instance, in "Harmless" tasks within the `Anthropic-HH` framework, responses beginning with phrases such as "Sorry," "No," or "I cannot help" can all validly express a preference to decline. Therefore, it is difficult to assert that the absence of a specific low-probability token necessarily results in a failure to align. In many cases, such a token can be effectively compensated for by other tokens or subsequent expressions.
>
> For **custom alignment tasks that do not primarily involve language expression**—such as those requiring the organization of content in rare formats or patterns—the general model may not have encountered these formats frequently, resulting in the low-probability issue you mentioned. In such cases, we could consider expanding the `Top-P/Top-K` selection range to include the target token. Alternatively, we could compute the probability distribution across the entire vocabulary (as outlined in Equation 9), incorporating adjustments from the Residual Aligner, and sample from that distribution, at the cost of higher computational expenses and potential instability from small-scale Residual Aligner adjustments.
>
> P.S. Your question has inspired us to consider methods for detecting the rarity of preferences by comparing the distributions of the Proposal Module and the Residual Aligner during both training and ongoing inference. We aim to incorporate this dynamic feedback into our `Top-P/Top-K` adjustments in future work.
>
> ---
>
> >W3: If my understanding is correct, one assumption here is that the Proposal Module and the Residual Aligner will share the same vocabulary space, which can lead to restricted application scenarios as well as the plug-n-play nature of the proposed pipeline. In this case, these two modules will need to come from the same model family of the same generation, so that they will be able to share the same vocabulary space. Authors also mention this limitation in their Appendix.
>
> Yes, it is indeed assumed that the Proposal Module and the Residual Aligner share the same vocabulary space, which we have mentioned in the Appendix. This is a common challenge in methods that involve logit ensembles or probability distribution fusion [1,2]. While this may present certain limitations, it is worth noting that there are numerous model families available, offering a range of sizes from small to large. Below is an incomplete summary of some prominent model families:
>
> | Model Family | Developer | Params |
> | --- | --- | --- |
> | Llama 3 | Meta | 1B, 3B, 8B, 70B, 405B |
> | Mistra l Mistral AI | 3B-124B |
> | Falcon 3 | TII | 1B, 3B, 7B, 10B |
> | Gemma 2 | Google | 2B, 9B, 27B |
> | Phi-3.x / 4 | Microsoft | 3.8B, 7B, 14B, 42B |
> | Command R | Cohere | 7B, 35B, 104B |
> | StableLM 2 | Stability AI | 1.6B, 3B, 12B |
> | Qwen2.5/3 | Alibaba | 0.5B-235B |
> | DeepSeek-R1 | DeepSeek AI | 8B-685B |
> | Yi | 01.AI | 6B, 9B, 34B |
>
> This diversity provides ample choices within open-source scenarios. However, challenges arise when attempting to combine modules from different model families. For example, using a closed-source LLM (such as GPT-4o) as the Proposal Module may create difficulties in finding a compatible open-source model that matches its vocabulary.
>
> Fortunately, recent efforts have been made to align the vocabularies of different models, which may help mitigate these limitations in the future [3-5].
>
> [1] Huang JY, Zhou W, Wang F, Morstatter F, Zhang S, Poon H, Chen M. Offset Unlearning for Large Language Models. TMLR'2025.
>
> [2] Mudgal S, Lee J, Ganapathy H, Li Y, Wang T, Huang Y, Chen Z, Cheng HT, Collins M, Strohman T, Chen J. Controlled Decoding from Language Models. ICML'2024.
>
> [3] Xu Y, Lu J, Zhang J. Bridging the Gap between Different Vocabularies for LLM Ensemble. NACCL'2024.
>
> [4] Huang Y, Feng X, Li B, Xiang Y, Wang H, Liu T, Qin B. Ensemble learning for heterogeneous large language models with deep parallel collaboration. NeurIPS'2024.
>
> [5] Wu J, Sun H, Cai H, Su L, Wang S, Yin D, Li X, Gao M. Cross-model control: Improving multiple large language models in one-time training. NeurIPS'2024.
>
> ---
>
> >W4-1: No statistical significance results are provided for the main results, but only for the ablation study.
>
> Due to budget constraints, we are unable to cover the high costs associated with calling GPT-4 to obtain statistical significance results. And, following the empirical approach outlined in paper *Aligner* (our baseline)  presented at NeurIPS 2024, we provide the win rates evaluated by the GPT-4 judger in our main experiments.
>
> >W4-2: Meanwhile, could authors also include a study in terms of the efficiency. For instance, what will be the running time and inference time overhead for adding the proposed method?
>
> To compare training efficiency, we use the Llama3 family as an example. The pre-trained 8B model has a forward cost of 1 computational unit and a backward cost of 2 units. For the small Residual Aligner (1B), the forward cost is 1/8 unit and the backward cost is 2/8 units. In SFT, the pre-trained model requires 3 computation units (1 forward, 1 backward), while the Residual Aligner needs only 6/8 units (2 forward, 2 backward for a paired example). This results in 4x increase in efficiency for SFT with the Residual Aligner.
>
> To compare inference efficiency, from the Equation 9, it can be observed that the Proposal Module $P_M$ and Residual Aligner $Q_\theta$ are loosely coupled, interacting only at the final logits layer (L147-L149). Therefore, during inference, the Proposal Module and Residual Aligner can be computed in parallel instead of serially, avoiding significant time overhead. Consequently, compared to the baseline model inference, RAM incurs only the additional time required for one pass of self-normalized importance sampling (L137).
>
> ---
>
> If our responses have contributed to a clearer understanding of our work and enhanced your appreciation for its value, we would greatly appreciate your consideration in updating your rating.

---

> > ### Comment · Reviewer_K98W · 2025-08-01
> >
> > I would like to thank the authors for their detailed rebuttal. I would like to keep my current positive rating of the paper, and encourage the authors to incorporate our discussions into the manuscript.

---

### Official Review · Reviewer_GBiF · 2025-07-02

**Clarity:** 3
**Significance:** 3
**Originality:** 2
**Rating:** 4
**Confidence:** 4

**Summary:**

This paper presents a clean and efficient framework for aligning large language models by treating the alignment process as importance sampling. The core idea is to keep the base model fixed and train a small residual module to steer the outputs toward preferred behaviors. The method is practical—it reduces training cost and improves inference speed—while maintaining solid performance across tasks. Although its generality is somewhat constrained and comparisons to stronger baselines are limited, the overall contribution is clear, timely, and useful.

**Questions:**

1.In the experimental section, the definition of the “warmed-up model” is unclear. Could the authors clarify the difference between a warmed-up model and an SFT model?

2.Compared to recent research that uses value functions for guided generation—also a form of weak-to-strong generalization—does your method offer any advantages? Could the authors elaborate on the relationship between your approach and these reinforcement learning-based methods?

3.Why did you choose token-level decoding? Wouldn’t this significantly increase decoding time? Have you considered chunk-level decoding or other alternatives? Just curious.

**Ethical Concerns:**

["NO or VERY MINOR ethics concerns only"]

**Final Justification:**

Thank you for your response. I believe my concerns have been addressed, and I’ve revised my score accordingly

**Limitations:**

yes

**Quality:**

3

**Strengths And Weaknesses:**

Strength:
1. The paper presents a smart and lightweight framework for aligning large language models by introducing a separate residual alignment module. This modular design allows the main model to remain frozen while only a small aligner is trained, leading to reduced computational cost and easier deployment.

2. Framing the alignment process as importance sampling is both elegant and practical. It offers a clean probabilistic interpretation of how the residual aligner adjusts the output distribution.

3. The paper is well writtened.

Weakness:

1. While the results look promising, the Residual Aligner (R.A.) appears to be heavily dependent on a strong proposal model. For example, in the HH experiments, although the 8B SFT + R.A. (3B or 1B) outperforms the SFT baseline, it still lags significantly behind the DPO method. Does this imply that the proposed method only works well when the base model is already strong? In that case, why not directly improve the base model instead?


2. The baselines used in the paper SFT and DPO are relatively outdated. I suggest including one or two additional recent baselines beyond Aligner to strengthen the empirical evaluation. For example, you might consider incorporating a controlled decoding approach such as:
[1] Mudgal, Sidharth, et al. "Controlled Decoding from Language Models." arXiv preprint arXiv:2310.17022 (2023).

---

> ### Author Rebuttal · Authors · 2025-07-31
>
> Thank you for your thoughtful review and for raising the above interesting questions.
>
> ---
>
> >W1-1: While the results look promising, the Residual Aligner (R.A.) appears to be heavily dependent on a strong proposal model. ... Does this imply that the proposed method only works well when the base model is already strong?
>
> We fully agree with your observation about the performance dependence of the Residual Aligner (R.A.) on the Proposal Module's strength. Our goal is to emphasize that **even a Residual Aligner with a limited number of parameters can enhance the performance of a strong Proposal Module**.
>
> Our experiments yield two main insights: first, as you pointed out, improving the Proposal Module is indeed valuable; second, a lightweight Residual Aligner can provide additional performance gains for already high-performing models.
>
> In our experimental design, we aimed to reflect real-world applications. Ideally, we would use a robust model like GPT-4o as the Proposal Module for constructing the Residual Aligner. However, challenges include the lack of lightweight models with the same vocabulary as GPT-4o and the unavailability of public high-quality datasets that surpass GPT-4o. When the Proposal Module (GPT-4o) surpasses the quality of available training data, training the Residual Aligner on "lower-quality" annotations becomes problematic. Additionally, experiments with non-public datasets encounter evaluability and reproducibility issues.
>
> Given these constraints, we opted for newer open-source models like Llama3.1-8B and Qwen2.5-14B as our Proposal Modules. We used pre-trained models to simulate scenarios **with adequate generalization but weak domain adaptation** and DPO models to represent **robust domain capabilities that still require enhancement**. This approach effectively simulates the model optimization challenges faced in real-world applications.
>
> >W1-2: In that case, why not directly improve the base model instead?
>
> Directly optimizing the base model for real-world applications faces critical pain points. Different downstream tasks often have customized requirements, necessitating the training of specific domain models based on the base model for each task. Beyond the training resource costs, **deploying separate domain models prevents the sharing of limited online service storage and computational resources, leading to significant inefficiencies**. In contrast, our method, utilizing a lightweight Residual Aligner, achieves domain adaptation or value alignment with fewer resources. **Multiple Residual Aligners targeting different domains can share the same Proposal Module, enabling efficient cross-domain resource reuse and enhancing the overall system's efficiency**. This principle is a core motivation of our work as mentioned in the Introduction and hope I have addressed your concern.
>
> ---
>
> >W2: The baselines used in the paper SFT and DPO are relatively outdated. I suggest including ... such as: [1] Mudgal, Sidharth, et al. "Controlled Decoding from Language Models." arXiv preprint arXiv:2310.17022 (2023).
>
> Thank you for pointing out the related works. We will include them in the Related Work section in our future paper updates. Due to time constraints, we tested and compared the performance of the *CD* method [1] on the *TL;DR* and *Anthropic-HH* datasets, with the evaluation results as follows:
>
> | TL;DR | LC/% | WR/% |
> | --- | --- | --- |
> | W.Up 8B | 60.71 | 49.02 |
> | W.Up 8B+Ali. 1B | 44.37 | 36.59 |
> | W.Up 8B+CD 1B | 61.89 | **52.90** |
> | W.Up 8B+R.A. 1B | **65.11** | 52.13 |
>
> | Helpfulness | LC/% | WR/% |
> | --- | --- | --- |
> | W.Up 8B | 57.70 | 56.60 |
> | W.Up 8B+Ali. 1B | 44.77 | 46.81 |
> | W.Up 8B+CD 1B | 58.08 | **59.25** |
> | W.Up 8B+R.A. 1B | **65.11** | 52.13 |
>
> | Harmlessness | LC/% | WR/% |
> | --- | --- | --- |
> | W.Up 8B | 66.63 | 64.88 |
> | W.Up 8B+Ali. 1B | 64.75 | 63.87 |
> | W.Up 8B+CD 1B | 67.00 | **66.73** |
> | W.Up 8B+R.A. 1B | **67.63** | 65.79 |
>
> Based on the results, we observed that our method outperforms the *CD* method in terms of the LC metric for both the *TL;DR* and *Harmlessness* tasks, showing significantly stronger performance in the *Helpfulness* task. However, it is worth noting that the *CD* method generally exceeds our approach in the WR metric. Within the AlpacaEval 2 evaluation framework, a lower LC metric with a higher WR metric suggests that the *CD* method tends to generate longer content. We summarize the average length of each strategy with comparison to that of the base prolicy in the table below:
>
> | avg_length | TL;DR | Helpfulness | Harmlessness |
> | --- | --- | --- | --- |
> | W.Up 8B | 115 | 230 | 134 |
> | W.Up 8B+CD 1B | 126 (+11) | 310 (+80) | 164 (+30) |
> | W.Up 8B+R.A. 1B | 112 (-3) | 247 (+17) | 129 (-5) |
>
> We speculate that the underlying issue stems from the use of value functions with fewer parameters for lightweight reweighting in *CD*. A significant concern is the direct influence of the energy function on the base policy (as outlined in Equation 6 of [1]), which can compromise its expressive capacity. This can lead to longer outputs or even result in model collapse when inappropriate hyperparameters are applied. And we will provide a detailed comparison of our method to CD in our response to "Q2".
>
> ---
>
> >Q1: In the experimental section, the definition of the “warmed-up model” is unclear. Could the authors clarify the difference between a warmed-up model and an SFT model?
>
> We appreciate your feedback about the ambiguity in the definition of the "warmed-up model", and will clarify this in future updates of the paper.
>
> In Table 3, the warmed-up model is trained on approximately 7-8% of the entire dataset. This phase begins with pretrained models and primarily focuses on adapting the model to recognize specific tokens introduced for conversational tasks, such as `<|start_header_id|>`, `<|end_header_id|>`, `<|eot_id|>`, etc., particularly within the Llama and Qwen model families. In comparison, the SFT model is trained on the complete dataset, which potentially include the warm-up process.
>
> Table 4 illustrates that the SFT model not only incorporates the functionalities of the warm-up phase but also addresses distribution shift issues that smaller models encounter during the DPO pre-training phase. This approach is designed to develop a robust base model capable of effectively generalizing to specific domains.
>
> ---
>
> >Q2: Compared to recent research that uses value functions for guided generation—also a form of weak-to-strong generalization—does your method offer any advantages? Could the authors elaborate on the relationship between your approach and these reinforcement learning-based methods?
>
> Taking [1] as a reference for comparison, the main differences lie in:
>
> | Strategy | Fundamental | Model Characteristics | Decoding Strategy |
> | --- | --- | --- | --- |
> | Value functions for guided generation (*CD*) | KL-regularized RL | Residual EBM [2] | Probability reweighting |
> | Ours | Importance sampling (IM) | Linear combination of auto-regressive LLMs | Self-normalized IM (SNIM) |
>
> Advantages of Our Method:
>
> - **Theoretical simplicity**: Our approach is theoretically straightforward, making it easier to understand and implement.
> - **Symmetrical modeling**: The model is structured as a linear combination of counterpart auto-regressive LLMs, which inherently supports the symmetry necessary for mutual importance weighting between LLMs. This foundation enables us to explore "speculative sampling" through chunk-level decoding, which proposes draft sampling conversely through the smaller Resisual Aligner and follows a smart rejection by Proposal Module—an avenue we plan to pursue in future work. In contrast, the Residual EBM [2] model, which combines LLMs with an energy function, lacks this extensibility.
> - **Mitigating degradation**: When employing value functions with fewer parameters for lightweight reweighting in *CD*, a significant concern arises: the direct influence of the energy function on the base policy (as outlined in Equation 6 of [1]) can compromise its expressive capacity. This may result in longer outputs or even lead to model collapse with inproper hyperparameters. In contrast, our approach, utilizing SNIM (referred to as *Proposing-Aligning-Reducing Sampling* in L137), allows for sampling from the `Top-P/Top-K` outputs of the base policy to prioritize basic fluency. This is followed by a secondary sampling step tailored to contextual alignment needs, effectively mitigating the risk of degradation. Furthermore, SNIM functions as a biased estimator with reduced variance, enhancing overall performance.
>
> Overall, our method not only addresses the limitations associated with reinforcement learning-based approaches [1] but also enhances the robustness and expressiveness of the generated outputs.
>
> [1] Mudgal, Sidharth, et al. "Controlled Decoding from Language Models." arXiv preprint arXiv:2310.17022 (2023)
>
> [2] Deng Y, Bakhtin A, Ott M, Szlam A, Ranzato MA. Residual Energy-Based Models for Text Generation. ICLR'2020.
>
> ---
>
> >Q3: Why did you choose token-level decoding? Wouldn't this significantly increase decoding time? Have you considered chunk-level decoding or other alternatives? Just curious.
>
> This paper primarily focuses on applying importance sampling to model lightweight domain adaptation, along with related training and inference methods, but not incorporated efficiency optimization into this study. However, as previously mentioned, the symmetry structure of our model supports the mutual importance weighting, allowsing us to explore "speculative sampling" through chunk-level decoding. We are already conducting related research works.
>
> ---
>
> If our responses have contributed to a clearer understanding of our work and enhanced your appreciation for its value, we would greatly appreciate your consideration in updating your rating.

---

> > ### Comment · Reviewer_GBiF · 2025-08-03
> >
> > Thank you for your response. I believe my concerns have been addressed, and I’ve revised my score accordingly

---

> > > ### Author Response · Authors · 2025-08-04
> > >
> > > Thank you for your recognition of our work! Your support means a lot to us. Wishing you all the best in your work and life!

---

### Official Review · Reviewer_523P · 2025-07-02

**Clarity:** 3
**Significance:** 2
**Originality:** 3
**Rating:** 4
**Confidence:** 3

**Summary:**

The paper presents a novel Residual Alignment Model (RAM) that reframes the task of aligning large pre-trained language models to human preferences as an importance-sampling problem. By treating the frozen base model as a “proposal” distribution and training a lightweight residual aligner to estimate the density ratio between the desired (biased) distribution and the proposal, the paper achieve efficient, modular alignment without full fine-tuning.
The paper also introduces a token-level “propose–align–reduce” decoding algorithm to avoid sequence-level latency and demonstrates its effectiveness on multiple benchmarks, showing that RAM matches or exceeds full-model fine-tuning and other baselines while using only a fraction of the trainable parameters.

**Questions:**

I have listed all of my concerns in the weaknesses section, and I look forward to the authors’ response.

**Ethical Concerns:**

["NO or VERY MINOR ethics concerns only"]

**Final Justification:**

The authors have addressed my concerns, and I believe the quality of the paper is acceptable. Therefore, I have raised my score to 4.

**Limitations:**

yes

**Paper Formatting Concerns:**

I have no Paper Formatting Concerns

**Quality:**

2

**Strengths And Weaknesses:**

Strengths:

1. By framing alignment as importance sampling, the paper achieves a clean decoupling of alignment from the base model. This modularity allows independent scaling and optimization of each component, and can also let multiple alignment modules share a single Proposal Module for cross-domain reuse;

2. The method yields a 4× speed-up for supervised fine-tuning (SFT) and a 13.3× speed-up for Direct Preference Optimization (DPO);

3. The paper conducts experiments across multiple instruction-following, summarization, and preference tasks using various model configurations, demonstrating that RAM matches or exceeds full-model fine-tuning and outperforms adapter-based Aligners and DPO baselines.

Weaknesses:

1. While the paper states that the new algorithm “minimizes first-word latency,” it only provides a rough estimate of computation units during training. It would be helpful to include real wall-clock measurements—such as latency curves, throughput numbers, or end-user response times—and to compare these against existing methods like Aligner, LoRA, or SFT.

2. The evaluation relies primarily on automatic judgments from GPT-4 and Qwen, without any human assessments to confirm consistency. I understand that rebuttal time may be limited, but including even a small human-rated subset (for example, a few dozen examples) could demonstrate alignment between the model’s automatic scores and human judgments.

3. The method still employs a modest amount of SFT “warm-up,” which seems somewhat at odds with the claim of “fully freezing the large model.” This suggests that, in resource-constrained environments, one must still initiate a LLM training pipeline. It might be useful for the authors to clarify this design choice or discuss its practical implications.

---

> ### Author Rebuttal · Authors · 2025-07-31
>
> Thank you for the insightful comments and helpful suggestions! We hope that our responses help alleviate the concerns that were raised.
>
> ---
>
> >W1: While the paper states that the new algorithm “minimizes first-word latency,” it only provides a rough estimate of computation units during training. It would be helpful to include real wall-clock measurements—such as latency curves, throughput numbers, or end-user response times—and to compare these against existing methods like Aligner, LoRA, or SFT.
>
> Thanks for highlighting the inaccuracy in our description about "minimizes first-word latency". Our intention was to clarify that, compared to the baseline Aligner's "Question-Answer-Correction" generation strategy, our method does not rely on a upstreaming complete response. This distinction allows us to significantly **reduce first-word latency**, enhancing the practicality of our method.
>
> We would like to clarify that this paper does not focus on the overall efficiency of the RAM framework during inference, which is why we have not conducted experimental validation in this area. To address your concerns, we have included performance testing experiments. However, due to time constraints, we primarily tested the SFT, Aligner, and RAM methods, while the performance of LoRA will be supplemented in future comments. The currently completed results are as follows:
>
> | Strategy | Avg Input Length (#tokens) | Avg Output Efficiency (tokens/s) | First Token Latency (s) |
> | --- | --- | --- | --- |
> | Llama8B (SFT) | 126 | 3.50 | 0.022 |
> | Llama8B+Ali. 1B | 126 | 5.03 | 50.72 |
> | Llama8B+R.A. 1B | 126 | 4.38 | 0.31 |
>
> It is worth noting that our method is structured as a linear combination of two autoregressive models $P_\theta(y_l|y_{<l},x)=\frac{P_M(y_l|y_{<l},x)Q_\theta(y_l|y_{<l},x)}{Z_\theta(y_{<l},x)}$ (refer to Equation 9). During the decoding phase, $P_M$ and $Q_\theta$ exhibit no dependencies, allowing for parallel processing at each iteration. Consequently, compared to the SFT model, RAM primarily incurs additional time for self-normalized importance sampling (referred to as *Proposing-Aligning-Reducing Sampling* in L137).
>
> Additionally, This linear combination structure mentioned above, which inherently supports the symmetry necessary for mutual importance weighting between LLMs. This foundation enables us to explore "speculative sampling" through chunk-level decoding, which proposes draft sampling conversely through the smaller Resisual Aligner and follows a smart rejection by Proposal Module—an avenue we plan to pursue in future work.
>
> ---
>
> >W2: The evaluation relies primarily on automatic judgments from GPT-4 and Qwen, without any human assessments to confirm consistency. I understand that rebuttal time may be limited, but including even a small human-rated subset (for example, a few dozen examples) could demonstrate alignment between the model’s automatic scores and human judgments.
>
> We apologize for the time constraints that limit our ability to conduct a comprehensive manual evaluation of the experimental results. Here, we randomly selected 50 samples labeled "Chosen" from the test set of the Harmlessness task in Anthropic-HH to assess the win rates of each *Strategy* against the "Chosen" samples, up to total 350 manual comparisons. The result is list below:
>
> | Strategy | GPT4-Eval | Human |
> | --- | --- | --- |
> | SFT 8B | 65.31 | 65.88 |
> | DPO 1B | 60.44 | 65.87 |
> | SFT 8B+Ali. 1B | 48.85 | 52.77 |
> | SFT 8B+R.A. 1B | 66.67 | 71.67 |
> | --- | --- | --- |
> | DPO 8B | 73.06 | 74.00 |
> | DPO 8B+Ali. 1B | 70.12 | 62.89 |
> | DPO 8B+R.A. 1B | 79.89 | 78.05 |
>
> There are discrepancies between the manual evaluations and GPT-4 Eval, primarily due to the limited sample size, which introduces significant variance. While it is difficult to definitively conclude whether manual evaluations align with LLM assessments, the manual evaluation trends indicate that the RAM strategy's relative advantage is likely preserved. We hope the following results address your concerns.
>
> ---
>
> >W3: The method still employs a modest amount of SFT “warm-up,” which seems somewhat at odds with the claim of “fully freezing the large model.” This suggests that, in resource-constrained environments, one must still initiate a LLM training pipeline. It might be useful for the authors to clarify this design choice or discuss its practical implications.
>
> We appreciate your feedback regarding the lack of context for the "warm-up" and will clarify this in future updates of the paper.
>
> We utilized the warm-up phase in experiments starting from the original pre-trained model. Pre-trained models often struggle with newly introduced special tokens (OOD tokens) in conversation tasks, such as `<|start_header_id|>`, `<|end_header_id|>`, `<|eot_id|>`, in the Llama and Qwen families. These tokens are integrated into conversation data during the "instruction-following" phase, allowing models to learn from them. Thus, our warm-up phase serves to prepare the model for understanding these special tokens.
>
> It is crucial to clarify that our method does not primarily address OOD issues. As stated in L67-L68, "The goal of alignment is to utilize instances from the biased subset $S$ to adapt the model $P_M$, aiming to make it a better estimator of $P_S$." Our objective is to align the model with the distribution of a specific domain subset, contingent on the proposal model's awareness of this distribution. This awareness supports our assertion that "$P_S$ is supported by $P_M$" (L78), ensuring that the importance weight $W=P_S/P_M$ remains bounded. The warm-up phase is essential for establishing this awareness, particularly concerning OOD token, allowing us to subsequently treat the proposal model as "fully frozen."
>
> Therefore, the warm-up phase is a necessary step for aligning pre-trained models in conversation tasks, rather than a contradictory condition.
>
> [If the above information suffices, please disregard the following.] To elaborate on our choice of starting with pre-trained models in our experiments:
>
> While a robust model like GPT-4o would ideally serve as the Proposal Module for the Residual Aligner, several challenges hinder this approach: To our knowledge, there is no lightweight, open-source alternative that matches GPT-4o's vocabulary. And the lack of widely-used datasets for alignment tasks that surpass the quality of GPT-4o complicates efforts to enhance its performance through training a Residual Aligner on these "lower-quality" annotations. Additionally, experiments based on non-public datasets face challenges regarding evaluability and reproducibility.
>
> Given these factors and aiming for a proof of concept, we opted for newer open-source models like Llama 3.1-8B and Qwen 2.5-14B, which can be fully parameter-trained within limited resources. We utilized pre-trained models to simulate scenarios with sufficient generalization capabilities but weak domain adaptation, alongside DPO models to represent robust domain capabilities still requiring enhancement. This approach allows us to effectively address the model optimization challenges encountered in real-world applications.
>
> ---
>
> If our responses have contributed to a clearer understanding of our work and enhanced your appreciation for its value, we would greatly appreciate your consideration in updating your rating.

---

> > ### Comment · Reviewer_523P · 2025-08-01
> >
> > Thank you for your reply. Your response has cleared up my doubts, and I’ve decided to raise the Rating to 4.

---

> > > ### Author Response · Authors · 2025-08-01
> > >
> > > Thank you for your recognition of our work. We truly appreciate your support! Wishing you all the best in your work and life.

---

### Note · Authors · 2025-08-14

We thank the ACs and reviewers for their thoughtful engagement. There is broad consensus on the unique contributions of this work:

- *523P*: "... achieves a clean decoupling of alignment ... allows independent scaling and optimization of each component, ... let multiple alignment modules share a single Proposal Module for cross-domain reuse."
- *GBiF*: "... Framing the alignment process as importance sampling is both elegant and practical. It offers a clean probabilistic interpretation ..."
- *K98W*: "... making LLM alignment affordable and efficient ... idea is intuitive ... a pragmatic and scalable approach for real-world use"
- *Nz81*: "... accelerates the inference process, ... The idea is simple and interesting"

The primary concern raised was regarding the potential high variance associated with importance sampling (by *K98W*) and comparisons with related RL-based approaches like CD (by *GBiF*).

**Mitigating Variance**: The RAM method incorporates a learnable Residual Aligner and normalizes to ensure $P_\theta$ is a valid probability distribution. This allows importance weights to be influenced by $Q_\theta$, correcting deviations between $P_M$ and $P_S$ and reducing variance. During inference, we use self-normalized importance sampling (SNIS) in the Proposing-Aligning-Reducing framework, sampling from the `Top-P/Top-K` regions of $P_M$. This maintains high $P_M$ values and lowers importance weights $W=P_S/P_M$, further mitigating variance via introducing a biased estimator.

**Advantages Compared to CD Method**: RAM provides theoretical simplicity over RL-based methods like CD. SNIS enhances fluency by sampling from the base policy's `Top-P/Top-K` outputs, reducing risks of degradation from smaller value functions in CD. Our linear combination of two auto-regressive LLMs facilitates mutual importance weighting and efficient speculative sampling through chunk-level decoding. Our experiments demonstrate RAM's advantages over CD, as detailed in our rebuttal to *GBiF*.

We will incorporate discussions with *523P* into the manuscript, highlighting the need for a "warm-up" step in the experimental setup for the pretrained model to learn the special tokens for conversational tasks. Additionally, we will include a comparison of first-word latency and inference efficiency in the Appendix.

These additions will enhance our original submission without requiring major changes. We appreciate the reviewers' valuable feedback, which has greatly refined our work.

---

### Decision · Program_Chairs · 2025-09-17

**Decision:**

Accept (poster)

**Comment:**

The paper presents a study on LLM alignment, and treat the alignment problem as a form of importance sampling. After effective rebuttal and discussion period, all reviewers turn positive to the paper. The reviewers mainly acknowledge the novelty of the Residue Alignment Model (RAM) which uses a separate and smaller module for alignment leaving the base model untouched, and the importance sampling treatment of the alignment problem. The paper meets the bar of NeurIPS and I would like to recommend accepting the paper. The authors need to carefully prepare the final version to address all constructive comments from the reviewers.

AC